# Regulatory ILC2—Role of IL-10 Producing ILC2 in Asthma

**DOI:** 10.3390/cells12212556

**Published:** 2023-10-31

**Authors:** Nahal Emami Fard, Maria Xiao, Roma Sehmi

**Affiliations:** Department of Medicine, McMaster University, Hamilton, ON L8N 3Z5, Canada; emamifan@mcmaster.ca (N.E.F.);

**Keywords:** interleukin-10, type 2 inflammation, allergic inflammatory disease, asthma

## Abstract

Over the past two decades, a growing body of evidence observations have shown group two innate lymphoid cells (ILC2) to be critical drivers of Type 2 (T2) inflammatory responses associated with allergic inflammatory conditions such as asthma. ILC2 releases copious amounts of pro-inflammatory T2 cytokines—interleukin (IL)-4, IL-5, IL-9, and IL-13. This review provides a comprehensive overview of the newly discovered regulatory subtype of ILC2 described in murine and human mucosal tissue and blood. These KLRG1+ILC2 have the capacity to produce the anti-inflammatory cytokine IL-10. Papers compiled in this review were based on queries of PubMed and Google Scholar for articles published from 2000 to 2023 using keywords “IL-10” and “ILC2”. Studies with topical relevance to IL-10 production by ILC2 were included. ILC2 responds to microenvironmental cues, including retinoic acid (RA), IL-2, IL-4, IL-10, and IL-33, as well as neuropeptide mediators such as neuromedin-U (NMU), prompting a shift towards IL-10 and away from T2 cytokine production. In contrast, TGF-β attenuates IL-10 production by ILC2. Immune regulation provided by IL-10+ILC2s holds potential significance for the management of T2 inflammatory conditions. The observation of context-specific cues that alter the phenotype of ILC warrants examining characteristics of ILC subsets to determine the extent of plasticity or whether the current classification of ILCs requires refinement.

## 1. Introduction

Innate lymphoid cells (ILC) are a family of lineage-negative cells that are comprised of natural killer (NK) cells, lymphoid tissue inducers (Lti), and three subgroups: ILC1, ILC2, and ILC3 (Figure 1) [1,2,3,4]. This nomenclature was proposed in 2013 [5]. Since their discovery, ILC have been recognized as primarily tissue-resident cells [6], particularly at the mucosal interface tissue of the airways, gastrointestinal tract, and skin [1,2], but ILC are also found in blood [6]. The estimates regarding the frequency of these cells put them at around 0.09% (0–1.5%) of total CD45+ cells in healthy individuals [7,8]. The earliest progenitor cells that give rise to the ILC family is the common lymphoid progenitor (CLP), which expresses an inhibitor of DNA binding 2 (Id2) [9]. Id2 is a fate-determining transcription factor that plays an indispensable role in the development of ILC as an inhibitor of E-protein transcription factors, which are important in adaptive immune cells [3,10]. CLP develop into early or common innate lymphoid progenitors (EILP/CILPs), which express transcription factor nuclear factor interleukin 3 (NFIL3) [11], thymocyte selection-associated HMG box (Tox), and maintain Id2 [12,13,14]. Natural killer progenitors (NKP) continue to express Id2, NFIL3 [11], and Tox, in addition to a transcription factor from the ETS family, ETS1, and give rise to mature NK cells [13,15,16]. Next, common helper innate lymphoid cell progenitor (CHILP) fails to generate NK cells but has been shown to give rise to all other ILC, including LTi cells (Figure 1) [10]. All CHILPs still express Id2 [13], but promyelocytic leukemia zinc finger (PLZF) containing CHILP also express GATA3 and give rise to groups ILC1-3, while PLZF lacking CHILP develop into LTi cells (Figure 1). CHILPs also lack expression of Tox, NFIL3, and ETS1 [10]. 

Functionally, ILC1-3 are classified as the lineage-negative cells’ innate immune counterparts of CD4+ T helper cells that lack the recombinase activation gene (RAG) and so do not respond in an antigen-selective manner. Like CD4+ T cells, ILC1 produce interferon-gamma (IFNγ) in response to IL-12, IL-15, and IL-18, which parallels T helper 1 (Th1) cells [17,18,19]. ILC2 produces several type 2 cytokines such as IL-5, IL-9, and IL-13 in response to thymic stromal lymphopoietin (TSLP), IL-25, and IL-33 and are analogous to type 2 T helper (Th2) cells [8,17,20]. Likewise, ILC3 are like T helper 17 (Th17) cells and mainly produce IL-17 and IL-22 in response to IL-23 and IL-1β (Figure 1) [17,21]. The expression of fate-determining transcription factors in ILC1-3 is also paralleled to their T cell counterparts. T-bet is mainly seen in ILC1 [14], while ILC2 and ILC3 express GATA3 [22,23,24] and RORγt [14,25], respectively (Figure 1). In terms of surface receptors, in both humans and mice, ILC1-3 are broadly defined as lin−CD94−CD45+CD127+ cells, meaning these cells lack the classic lineage markers that T and B cells have, which generally include CD3+CD14+CD16+CD19+CD20+CD56+FcER1+ cells; hence, they are known as innate cells as opposed to adaptive cells [26]. Human ILC1 are further defined as CD103+CRTH2−NKp44−/+CD117− cells [27], while human ILC2 are broadly described as ST2+CD25+CysLTR1+CRTH2+KLRG1+/− cells [28] and ILC3 are generally CRTH2–CD117+NKp44+/− [26,29]. 

NK cells and LTis are included in this family as they share features with ILC1 and ILC3, respectively, in development, phenotype, and function (Figure 1). While ILC1 and NK cells are primarily activated in response to viruses, intracellular microbes, and cancers, ILC2 are implicated in the defense against helminth [30] and viral infections [31], as well as having a role in allergic diseases [32,33]. Thus far, ILC2 have been implicated in several allergic diseases such as asthma [20], chronic rhinosinusitis [34], atopic dermatitis [35,36], nasal polyposis [34,37], and food allergy [38,39,40], and as having a central role in driving Type 2 (T2) inflammation. ILC3 are proposed to drive the recruitment of neutrophils through the Th17 immunity pathway in response to epithelial tissue alarmins released during extracellular microbes or fungi infections. The IL-17 produced by ILC3 may have a role in chronic obstructive pulmonary disease (COPD) [41] and driving neutrophilic bronchitis in asthma, characterized by >64% neutrophils in sputum differential cell counts [42]. The focus of this review will be on immunobiology of ILC2. 

## 2. Asthma and Type 2 Inflammation

The immunological response triggered by allergens generally triggers a Type 2 (T2) inflammatory cascade (Figure 2). In allergic asthma, atopic dermatitis, and allergic rhinitis, otherwise innocuous allergens, including grass or tree pollens, house-dust mites, cockroaches, or animal dander will stimulate the classic adaptive immune pathway to produce IgE directly specifically against the allergen [43]. This process is orchestrated by the maturation of CD4+ Th2 cells in lymph nodes and the subsequent production and release of T2 cytokines by these cells, such as IL-4, IL-9, IL-5, and IL-13, within tissues [43]. In contrast, ILC responds to microenvironmental cues from epithelial and other immune cells as opposed to direct allergen presentation in an innate immunity path (Figure 2). Discovered almost simultaneously in 2010 by Moro et al. [44], Neill et al. [45], and Price et al. [46], ILC2 cells feature prominently as effector cells of airway inflammation in asthma. Epithelial cells in mucosal tissues can be damaged by the protease activity of allergens or environmental elements such as pollution and viral infection, triggering alarmin production, including TSLP, IL-33, and IL-25 [43,47]. These alarmins prompt ILC2 to secrete T2 cytokines. Additionally, lipid mediators such as PGD2 and Cysteinyl leukotrienes (CysLTs) stimulate ILC2 [42,48]. Overall, T2 cytokines prompt the features of allergic diseases, which in the context of asthma is a substantial influx of eosinophils to mucosal sites [49,50]. Eosinophils also release T2 cytokines as well as cytotoxic granules that damage the integrity of the epithelial barrier, leading to a greater release of alarmins, thereby driving a chronic inflammatory response [51]. Airway eosinophilia is characterized by >3% eosinophils in sputum [52]. Processes triggered by T2 cytokine expression ultimately lead to mucus hypersecretion in the airways, variable airflow obstruction, and airway/bronchial hyperresponsiveness [51,53], which contribute to symptoms of asthma such as coughing, wheezing, and shortness of breath [53,54,55]. Consequently, the identification of regulatory mechanisms that dampen this vicious feedforward loop is crucial for the management of T2 inflammatory disease.

Although ILC2 are numerically fewer than CD4+ Th2 cells in asthmatic lungs (10,000-fold fewer), the cell produces up to 100-fold more T2 cytokines than the latter on a cell-per-cell basis and likely contributes to the persistence of eosinophilia in the airways [8,20,56]. Additionally, ILC2 may be the main driver of eosinophilia and asthma symptoms in conditions where CD4+ T cell activity has been dampened by steroid therapy recapitulated as knocking out recombinase activating gene (*Rag^−/−^*) in mice models [47,57,58]. Additionally, ILC2 are significantly increased in the blood of patients with severe (7844 ± 2350 cells/mL) compared to mild (310 ± 74 cell/mL) asthma and similarly in the sputum of severe (8715 ± 2404 cells/mL) vs. mild (182 ± 44 cells/mL) asthma patients [8,59], where ILC2 levels correlate with disease severity [55,60,61,62]. It is now postulated that the co-operative interaction of CD4+ Th2 and ILC2 function to drive inflammation in allergic respiratory diseases [63].

Importantly, it has recently been realized that considering only T2 inflammation in asthma is an oversimplification, and asthma endotypes are now broadly categorized as the classic T2 high and a T2 low endotype that presents through Th1 and Th17 immune pathways [43]. T2 low asthma is often accompanied by neutrophilia (>64% neutrophils in sputum differential cell counts), mainly driven by IL-17, as opposed to eosinophilia [43]. There is also a mixed granulocytic endotype of asthma, which is the simultaneous presence of eosinophilia and neutrophilia, and the paucigranuloytic endotype, characterized by <3% eosinophils and <64% neutrophils in sputum differential cell counts [64,65]. This complex landscape underscores the multitude of innate and adaptive immune cells and interactions within the tissue microenvironment that collectively contribute to the pathogenesis of asthma. Therefore, understanding and harnessing the body’s intrinsic regulatory pathways responsible for controlling inflammation is paramount to the management of allergic diseases such as asthma. 

## 3. Plasticity of ILC

ILCs were initially considered to be terminally differentiated cells, distinguished by the receptors, transcription factors, and panel of cytokines listed above. However, there has been a paradigm shift based on recent evidence that ILC are plastic. These cells can transdifferentiate and inter-convert between the defined subgroups and can produce a spectrum of overlapping cytokine profiles depending on signals from the local tissue milieu [66]. For example, Cai et al. [67] found that through induction with IL-25, IL-33, and lipid mediators, including cysteinyl leukotrienes (CysLT) and prostaglandin D4 (PGD4), mice lung ILC2 can be induced to produce IL-17 thereby shifting to display an ILC3-like profile. Other studies have implicated cues such as IL-1β, IL-23, and TGF-β in driving the plasticity of ILC2 to ILC3 [68,69]. More recently, these ILC-3, like ILC2, which are CD117+ and IL-17 producing, have been shown to contribute to Th17-mediated pathologies and are implicated in causing neutrophilic inflammation in airway diseases [67,68,69,70,71]. 

Plasticity has also been reported among ILC1 and ILC3 in the human intestines, especially in relation to Crohn’s disease, where there is a marked decline in ILC3 in favor of ILC1, while healthy individuals have more ILC3 and fewer ILC1 [72]. Tissue microenvironmental cues are driving this trans-differentiation from ILC3 towards ILC1, specifically through IL-12 and IL-23 [73,74]. Furthermore, in vitro studies have shown that ILC2 and ILC3 shift towards an ILC1 phenotype under the influence of IL-1β and IL-12 [75,76]. In vivo, conversion of ILC2 to ILC1 occurred in the lungs of mice infected with influenza A virus, which led to a rapid downregulation of GATA3, characteristic of ILC2 [76]. Meanwhile, IL-4 seems to be critical in maintaining the typical ILC2 phenotype since it was able to reverse the conversion of ILC3-like ILC2 back toward T2 cytokine production [68,70]. ILC2 to ILC1 trans-differentiation also appears to be reversed in the presence of IL-4, which favors the ILC2 phenotype by activating the STAT6 pathway, leading to the expression of GATA3 and CRTH2 [69,77]. 

Of specific interest to this paper are recent studies reporting a novel subtype of regulatory CRTH2+/−KLRG1+ILC2 that express the anti-inflammatory cytokine IL-10, which downregulates T2 cytokine production and decreases eosinophil recruitment [78,79]. Furthermore, the presence of IL-10+ILC2 had clinical benefits in inflammatory and allergic conditions such as grass pollen allergy [80,81,82]. These findings demonstrate plasticity within the major ILC subgroups and posit that within each subset, ILC produces variable cytokine profiles according to local microenvironmental prompts, which in turn contribute to immune regulation or dysregulation.

## 4. Interleukin-10

Interleukin (IL)-10 is an important anti-inflammatory or immune-regulatory cytokine that maintains homeostasis and prevents tissue damage due to immune hyperreactivity [83]. IL-10 binds to a heterodimeric receptor and activates the JAK1/STAT3 pathway, promoting upregulation of itself, along with anti-inflammatory molecules such as suppressors of cytokine signaling 3 (SOCS-3) [84,85]. The overall effect of IL-10 is the downregulation of cytokine production and pro-inflammatory genes in their targets T cells, B cells, NK cells, and granulocytes, which in turn may downregulate airway inflammatory responses as seen in asthma [79,84]. Dysfunction or dysregulation of IL-10 contributes to numerous human autoimmune diseases and allergic conditions [84,86,87]. Consequences of impaired IL-10 function in experimental disease models include chronic inflammatory bowel disease, rheumatoid arthritis, psoriasis, lupus, multiple sclerosis, transplant rejection, cancer, and infection [86]. Moreover, mutations of *IL10* that decrease IL-10 production are correlated with the incidence of allergy [87]. Regulatory T (Treg) [88] and B (Breg) cells are known to be primary producers of immunoregulatory cytokines such as IL-10 and TGF-β [1,79,81]. In addition, a newly described subset of ILC2 that contributes to homeostasis at mucosal sites demonstrates the autologous production and secretion of IL-10 [79,81,89]. Given its wide array of targets and its spectrum of anti-inflammatory properties, therapeutic manipulation to increase endogenous IL-10 production by various immune cells is the focus of several human disease conditions [90,91]. 

## 5. The Role of IL-10 Expressing ILC2

Seehus et al. [79] were the first to identify this as a subclass of ILC2 capable of IL-10 production. These cells have regulatory functions but do not express Foxp3 [92], making them molecularly and phenotypically distinct from Tregs [79]. Additionally, it has been noted that ILC2 produces IL-10 more readily than ILC1s and ILC3s, supporting the existence of IL-10+ILC2 as a distinct sub-class [2]. IL-10+ILC2 are characterized by their high expression of the transcription factor Id3 and tend to express surface receptor killer cell lectin-like receptor G1 (KLRG1) [2,79]. Here, we will investigate emerging evidence that resident and circulating naive ILC2 recruited to mucosal sites can differentiate into mature ILC2 that steadily produce IL-10 in response to molecules such as IL-2, IL-4, IL-10, IL-33, neuromedin-u (NMU) and retinoic acid (RA) (Table 1).

Firstly, IL-10+ILC2 has been detected in the lungs of healthy mice after chronic treatment with the allergen papain or IL-33 [79,93]. In humans, IL-10+ILC2 were rarer in individuals with grass pollen and house dust mite allergies compared to healthy individuals, indicating that these cells could facilitate tolerance to allergens [81]. A recent study assessed IL-10+ILC2 in regulating grass-pollen-induced inflammatory responses in sensitized mice [81]. It was found that IL-10+ILC2 attenuated the T2 response and maintained the integrity of the epithelial barrier, effectively inhibiting the main drivers of eosinophilic bronchitis [81]. Furthermore, using murine models of asthma, Howard et al. [80] observed the regulatory role of IL-10+ILC2 in dampening airway hyperresponsiveness and eosinophilia in the lungs. As such, perhaps the airway hyperresponsiveness and bronchial hyperinflammation in asthma may be attributable to inadequate downregulation of immune responses due to diminished numbers of IL-10+ILC2. Thus, it would also be expected that IL-10+ILC2 levels are increased in atopic subjects following successful allergen immunotherapy (AIT) [81]. A different study found that AIT can induce long-term immune tolerance through induction of regulatory T cells, regulatory B cells, and inhibitory cytokines such as IL-10 and TGF-β, promotion of immunoglobulin class switching, and inhibition of Th2 inflammatory response [94]. Furthermore, the frequency of type 2 innate lymphoid cells (ILC2) was downregulated, but the number of IL-10-producing ILC2 in patients increased by AIT [94]. In line with this, in a recent prospective, double-blind, placebo-controlled study, Golebski et al. [81] found that sublingual and subcutaneous AIT for grass pollen allergy increased anti-inflammatory IL-10+ILC2 to levels equivalent to those found in non-atopic individuals [81,95]. The frequency of IL-10+ILC2 was found to increase with the improvement of clinical symptoms in atopic subjects. Similar to previous findings, children with moderate and steroid-resistant asthma have been found to have significantly diminished levels of IL-10 in their airway lavages and peripheral blood mononuclear cells (PBMCs) when compared to non-asthmatic controls [96]. This further suggests a defective IL-10 system in asthma and calls for therapeutic methods that can raise IL-10 levels to healthy levels. Interestingly, dexamethasone treatment increased IL-10 in the PBMCs of these children in vivo as well as in vitro, while the addition of vitamin D3 (1α,25-dihydroxyvitamin D3) in culture further enhanced dexamethasone-induced IL-10 [96]. Additionally, in a cohort control study that examined AIT efficacy in individuals with allergic rhinitis in response to house dust mite (HDM), Boonpiyathad et al. [95] found that the percentage of IL-10+CTLA-4+ILC in individuals who responded to treatment was increased by 3.2% (95% CI) after 2 years. Concurrently, IL-10+ILC2 were greater in individuals who responded to treatment than non-responders and those in the placebo control group [95]. Levels of IL-10+ILC2 cells in mice declined within 14 days following the stoppage of stimulation with an allergen or IL-33 challenge [79]. However, a small number of these cells were maintained and quickly re-mobilized upon one restimulation injection with IL-33, as late as 30 days post-initial allergen challenge. This shows that ILC2 can be permanently polarized into IL-10+ILC2 and maintained to rapidly respond to subsequent allergen exposure, analogous to T memory cells. Together, these findings suggest that IL-10+ILC2 have an essential influence on tolerance to allergens and may be a good biomarker to assess AIT effectiveness.

Using IL-10 in a therapeutic context, when bleomycin-induced pulmonary fibrosis model mice were given IL-10 gene delivery treatment through intravenous injection of Ringer’s solution containing IL-10 plasmid, the pathological findings were significantly reduced along with TGF-β_1_ in bronchoalveolar lavage fluid (BALF) [97]. IL-10 also specifically suppressed α_V_β_6_ integrin on lung epithelial cells, which is important to the TGF-β_1_ pathway. Further, it was found that alveolar macrophages from bleomycin-injected mice produced TGF-β_1_ spontaneously ex vivo, which was significantly suppressed by treating the mice with IL-10 gene delivery in vivo or by treatment of the explanted macrophages ex vivo with IL-10 [97]. 

## 6. Positive Modulators of IL-10 Expression by ILC2

Retinoic acid (RA) is a potent inducer of IL-10+ILC2 (Figure 3). In 2021, Golebski et al. [81] categorized ILC2 isolated from human peripheral blood into four groups including KLRG1+CRTH2+, KLRG1−CRTH2+, KLRG1+CRTH2–, and KLRG1−CRTH2–, and analyzed these in cell cultures. The data showed that KLRG1+CRTH2−ILC2 produced the greatest quantity of IL-10 following in vitro incubation with IL-2, IL-7, and IL-33 in the presence of RA [81]. IL-7 was included as a growth factor for ILC2 to maintain the viability of cell cultures and is not directly correlated with IL-10 production. The production of IL-5 and IL-13 was detected in all conditions, while only ILC2 cultured in the presence of RA produced IL-10, but not ILC2 stimulated with IL-2, IL-7, and IL-33 alone [81]. Therefore, both in combination with IL-2 and IL-33, as well as alone, RA is a powerful stimulus of IL-10 production by ILC2 in a dose-dependent manner [79,81]. Other studies have also found RA to be necessary for the conversion of inflammatory ILC2 to IL-10+ILC2 [95,98]. One distinction between ILC2 and the Th2 counterparts is the presence of retinoid-related orphan receptor alpha (RORα) in ILC2 [99]. RORα is critical for ILC2 development but has a limited role in Th2 cells [42]. RA itself is a vitamin A metabolite, and CD103+ dendritic cells (DCs) are a major source of RA within the lung [100]. Mice that are deficient in CD103+DCs mimicking an RA-deficient microenvironment showed a reduction in the *IL10* gene expression by ILC2 previously activated in vivo with IL-33 injections [79].

As discussed, epithelial cell-derived IL-33 is a potent inducer of the proliferation and differentiation in ILC2 and induces T2 cytokine production by these cells. However, one study reported that exposure to IL-33 (four daily injections) and chronic exposure to allergen papain (five daily injections, one week rest, then five more daily injections) caused the blood ILC2s of mice to secrete IL-10 [79]. Meanwhile, long-term clonal expansion experiments showed that once KLRG1+ILC2 are polarized toward IL-10 production, they continue to produce IL-10 upon activation with IL-33 and TSLP alone, even when RA was subsequently absent [81]. In addition, after IL-10-GFP (green fluorescent protein) mice were given 3 days of intranasal IL-33 to activate ILC2, culturing the isolated ILC2 for just 2 days with IL-2, IL-4, and IL-7 led to about half the total ILC2 population to become IL-10+ [80], measured as a function of positive GFP fluorescence. These IL-10+ILC2 maintained IL-5 production but had attenuated IL-13 production as compared to ILC2 only activated with IL-33 [80]. Therefore, epithelial-derived alarmins IL-33 and TSLP can be potential enhancers of previously polarized IL-10+ILC2 but do not seem sufficient inducers of IL-10 production [79]. 

Contrastingly, IL-2 is another inducer of IL-10+ILC2 generation (Figure 3). The administration of IL-2 to healthy mice was effective at inhibiting eosinophilia in the lungs, which was associated with increased IL-10+ILC2 numbers, even in the absence of adaptive immune cells [79]. Some production of IL-5 and IL-13 from these IL-10+ILC2 was maintained [79]. In addition, the incubation of IL-10+ILC2 with IL-2 in culture has been shown to increase the number of IL-10+ILC2 and expand the secretion of IL-10 on a per cell basis [21,79]. Interestingly, IL-10 production was optimally induced in culture by the addition of IL-2 and RA, as compared to IL-2 or RA alone [79]. Moreover, in the brain meninges of both WT and Rag1 deficient mice who lack adaptive immunity, intracisternal administration of IL-33, which activates the AMP-activated protein kinase (AMPK), led to suppressed ILC1 and ILC3, while IL-10+ILC2 numbers increased [101]. There is some evidence that combined stimulation of ILC2 with IL-2 and RA indirectly exerts an effect by increasing the efficiency of IL-33 to induce IL-10 production [2,98], but this requires further investigation. The potential source of IL-2 in vivo is activated lung mast cells [95]. Seehus et al. [79] noted that mice deficient in mast cells had reduced IL-10 secretion capacity by ILC2, pointing to a role for mast-cell-derived IL-2 in IL-10+ILC2 activation.

The significance of IL-2 for IL-10+ILC2 was confirmed by Bando et al. [21], who also showed that stimulation with IL-4, IL-10, and neuropeptide NMU strongly induced IL-10 production by intestinal ILC2 in mice. In addition to this, cultures using specific blocking antibodies revealed that IL-10 production was effectively stimulated by both IL-2 and IL-4 independently [21]. Furthermore, ILC2 expresses the IL-10 receptor and, when stimulated with IL-10, shows a marked reduction in T2 cytokine production, namely IL-5 and IL-13 [7,21]. IL-10 secretion itself further induced IL-10 production by ILC2 through an autocrine effect [21,79,80]. In support of this, ILC2 cultures with IL-10 inducers (IL-2 and IL-4) treated with IL-10-blocking antibodies showed that IL-10-eGFP expression was blunted by inhibiting IL-10 [21]. 

A different study by Howard et al. [80] provided further support that IL-4 alone can induce IL-10+ILC2 (Figure 3). Supernatants from culturing IL-33-activated ILC2 with IL-4-producing Th2 cells demonstrated a dramatic increase in the level of IL-10, which was significantly reduced when an anti-IL-4 antibody was added [80]. Because Th2 cells are dominant sources of IL-4 in lung inflammation, Th2 cell-derived IL-4 may be important for stimulating IL-10 production by ILC2. However, Th2-derived IL-4 is also necessary for the accumulation of inflammatory ILC2 within the lungs, and IL-4 produced by ILC2 is important to Th2 cell development and function (Figure 2) [102,103,104]. Thus, like IL-2, it is not exactly clear in which contexts IL-4 would stimulate more T2 cytokine or IL-10 production. Notably, IL-4 stimulation can induce IL-10 production in other cell types, namely Th1 cells. In vitro, antibodies blocking IL-4 inhibited IL-10 production by Th1, while the addition of exogenous IL-4 to Th1 enhanced IL-10 [105]. Furthermore, IL-4 derived from Th2 cells, specifically, have been shown to induce IL-10 production by Th1 cells in culture [105]. Zhu et al. [106] have shown that when DCs are stimulated with IL-4 and RA, a shift from a pro-inflammatory to an anti-inflammatory profile of cytokines is observed. This, in conjunction with the recognized significance of RA to IL-10 production by ILC2, suggests that the combination of IL-4 and/or IL-2 with RA in the local milieu leads to the induction of an immune regulatory profile of ILC2. 

**Figure 3 cells-12-02556-f003:**
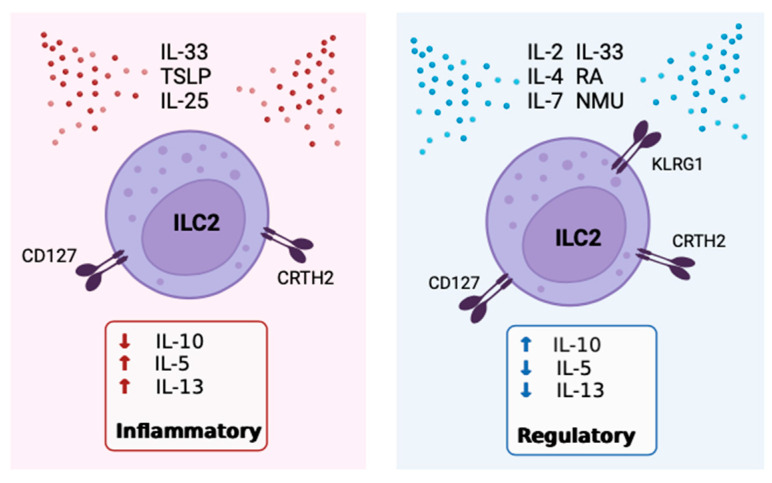
Inflammatory vs. regulatory ILC2 main inducers, characteristic receptors, and cytokine products. The figure details the currently known stimuli that promote either the regulatory or inflammatory phenotype of ILC2.

## 7. Inhibitors of IL-10 Expression by ILC2

The multitude of stimulating factors indicates that there is a complex interaction between cells within the local tissue that supports IL-10+ILC2 differentiation and proliferation. Conversely, tumor necrosis factor-like peptide 1A (TL1A), a member of the tumor necrosis factor superfamily (TNFSF15), has been found to suppress IL-10 production by ILC2 stimulated with IL-2 or IL-4 in culture [21]. Human ILC2 mainly express CRTH2, CD127, ST2, and CD25 [17] but also express death receptor 3 (DR3), a member of the TNF-receptor superfamily (TNFRSF25), which is a receptor specific to TL1A [107]. The TL1A/DR3 axis activates conventional ILC2, contributing to the optimization of T2 responses [107]. TL1A has been specifically shown to induce the production of IL-5 and IL-13 by ILC2 and promote allergic inflammation [2,108]. Alongside TL1A, experiments adding TGF-β to ILC2 isolated from peripheral blood of subjects with systemic sclerosis that were cultured on dermal fibroblast showed decreased IL-10 production and dramatically decreased KLRG1 expression on ILC2, associated with IL-10 generation [109]. The addition of TGF-β also attenuated IL-10 production by lung ILC2 that was evoked by IL-2 and RA in culture [79].

## 8. Receptors, Sex Disparities and Neuroimmune Control of IL-10+ILC2

In examining surface markers, Golebski et al. [81] found that ILC2 in blood expressing KLRG1+ but not KLRG1- produced IL-10, regardless of CRTH2 expression. KLRG1, or killer cell lectin-like receptor G1, is a co-inhibitory receptor that binds its ligand E-cadherin with low affinity [110]. E-cadherin is expressed by epithelial cells (ECs), dendritic cells (DCs), and other antigen-presenting cells [111]. Bando et al. [21] also noted that only KLRG1+ILC2 were capable of IL-10 production, while no other lineage-negative cells produced IL-10. In systemic sclerosis, skin biopsies show increased overall ILC2 numbers compared to healthy donors, and KLRG1-ILC2 specifically were elevated compared to KLRG1+ILC2 [109]. The percentage of KLRG1+ILC2 was negatively correlated to the extent of cutaneous fibrosis in these patients, implicating this surface receptor as indicative of the anti-inflammatory properties of ILC2 [109]. This also points towards ILC2 plasticity, as it signifies that CRTH2+ILC2 with a pro-inflammatory cytokine profile may be modulated by the local tissue microenvironmental cues to develop an anti-inflammatory property associated with increased KLRG1 expression and IL-10 secretion. 

Notably, recent studies [112,113,114] have also pointed toward the presence of KLRG1 as one of the main drivers of phenotypic differences in ILC2 that contribute to sex disparities in asthma where prevalence is greater in females compared to males, post-puberty. Growing evidence indicates that ILC2 from males express more KLRG1 than ILC2 from females [112,113,114,115], while females generally have more ILC2 than males. Additionally, 5alpha-DHT or testosterone can attenuate pro-inflammatory T2 cytokine secretion by ILC2 [116]. Specifically, male lung ILC2 stimulated with IL-2 and IL-33 showed decreased IL-5 and IL-13 production by ILC2 compared to female lung ILC2, proposed to be mediated by testosterone [116]. Kadel et al. [115] identified a subset of KLRG1-ILC2 in the lungs of female mice as the major contributor to sex bias in ILC2 numbers and disease. Also, they found that elevated androgens led to decreased cytokine production by lung ILC2 in both sexes of mice, while removal of estrogen did not have a major effect on ILC2 [115]. Thus, testosterone potentially has an immune-protective effect by promoting KLRG1+IL-10+ILC2. To date, the effects of KLRG1 on ILC2 in driving sex differences have only been studied in mouse models. Importantly, only KLRG1 was associated with the sex disparities, and whether the KLRG1+ILC2 cells produce IL-10 and the association of IL-10+KLRG1+ILC2 and sex differences in asthma has not yet been investigated. Human studies are needed to evaluate the relevance of KLRG1+ and KLRG1-ILC2 subsets and IL-10 production in asthma pathophysiology and sex disparity. 

Apart from KLRG1, studies have implicated cytotoxic T lymphocyte-associated protein 4 (CTLA4) and CD25 expression with the production of IL-10 by ILC2 post-AIT [95,98]. The Ctla4 gene was also upregulated in IL-10+ILC2 [79]. Conversely, the surface intercellular adhesion molecule ICAM-1 has been shown to play a particularly important role in the development and function of pulmonary ILC2 [117] and has been implicated in ILC2-dependent airway hyperreactivity and inflammation in mice [118]. Impairment in the ERK signaling pathway in ICAM-1-/-ILC2 led to an upregulation of IL-10 production and a decline in GATA3 levels in these cells [2,118], which in turn reduced output of T2 cytokines (IL-5, IL-9, and IL-13) upon allergen challenge [2]. Furthermore, the presence of ICAM-1 directly inhibited IL-10 production in murine lung ILC2, and anti-ICAM-1 treatment downregulated IL-5 and IL-13 secretion whilst triggering greater IL-10 expression [2,116,118]. This effect was also observed in ILC2 from T- and B-cell deficient mice [118]. Therefore, it is possible that IL-10-producing ILC2 do not express ICAM-1 on its surface like conventional ILC2, although this remains to be investigated in humans.

The neuropeptide, neuromedin U (NMU), which is the ligand of NMUR1, has demonstrated the capacity to rapidly activate ILC2 and presents an important neuroimmune axis in inflammatory conditions [89,119]. Also, in vivo co-administration of NMU with IL-25 strongly amplified allergic inflammation [89]. However, Bando et al. [21] found NMU to induce IL-10 production in murine intestinal ILC2 in culture experiments. On the other hand, semaphorin (Sema6D) is an axonal guidance cue that is produced by mesenchymal cells, and lung ILC2 from Sema6d−/− mice showed increased KLRG1 and PD-1 expression. All the while, Sema6D deficiency did not affect Id2 or GATA3 expression in ILC2 [120]. Also, ILC2 from the lungs of Sema6d−/− mice cultured with IL-2 and IL-33 exhibited significantly decreased IL-5 and IL-13 but augmented IL-10 synthesis compared to WT mice, as seen through cytokine and mRNA analysis [120]. Concurrently, Sema6d−/− mice showed significantly diminished eosinophilia and ILC2 numbers in bronchoalveolar lavage fluid (BALF) after challenge with IL-33, as well as elevated IL-10 and less IL-5 and IL-13 in BALF compared to WT mice [120]. Extended inquiry to clarify the function of NMU and Sema6d in relation to IL-10+ILC2 is warranted.

**Table 1 cells-12-02556-t001:** Summary of the inducers and inhibitors of IL-10 production by ILC2 that have been identified in the included studies.

Study	Models and Methods	Inducers of IL-10	Inhibitors of IL-10
Bando J. et al. [21]	ILC2 from small intestine lamina propria of WT mice subjected to intestinal inflammation.	IL-2, IL-4, IL-10, IL-27, and NMU	TL1A
Seehus C. et al. [79]	WT murine blood and lung ILC2, pre-activated +/− intraperitoneal IL-33.	IL-2, IL-33, RA	TGF-β
Howard E. et al. [80]	Blood ILC2 from WT and RAG deficient (*Rag2^−/−^γc^−/−^*) in a mouse model of asthma.	IL-2, IL-4, cMaf, Blimp-1	-
Golebski K. et al. [81]	Murine lung-tissue-extracted ILC2 using a model for grass pollen allergic rhinitis. Blood ILC2 from healthy controls and grass pollen allergic rhinitis.	IL-2, IL-33, RA, KLRG1	-
Wang S. et al. [82]	Intestinal ILC2 from healthy and *Rag1^−/−^Il10^−/−^* mice and human intestinal tissue.	TGF-β	-
Boonpiyathad T. et al. [95]	PBMC from subjects with allergic rhinitis undergoing AIT.	RA, CD25, CTLA-4	-
Baban B. et al. [101]	Examined CSF of traumatic brain injury patients and model mice and analyzed the effect of metformin.	KLRG1, IL-2, IL-33, AMPK	-
Laurent P. et al. [109]	Skin biopsies and peripheral blood of systemic sclerosis patients.	KLRG1	TGF-β
Hurrell BP. et al. [118]	Asthma model mice and ICAM-1 deficient mice challenged intranasally with IL-33.	-	ICAM-1
Naito M. et al. [120]	Isolated lung ILC2 from WT, allergic lung inflammation model mice, and Sema6D deficient mice.	-	Sema6D
Morita H. et al. [121]	Isolated ILC2 from the blood and nasal tissue of healthy and chronic rhinosinusitis subjects and HDM-sensitized mice.	IL-2, IL-33, RA, CTLA-4	TGF-β

## 9. Mechanisms Associated with IL-10 Expressing ILC2

Although IL-10+ILC2 cells are found in several tissues, the precise cellular mechanisms that enhance IL-10 synthesis by ILC2 remain unclear. For example, it is known that activated ILC2 utilize fatty acid oxidation for fuel and that their metabolic activity is controlled by the presence of fatty acids in the cells’ microenvironment [80]. Remarkably, it was observed that IL-4 stimulated IL-10+ILC2 demonstrated a switch towards the glycolytic pathway and shifted away from the traditional fatty acid oxidation in ILC2 that promotes pro-inflammatory functions [80]. Furthermore, in a study with subjects who have traumatic brain injury (TBI), ILCs were found to be elevated in cerebrospinal fluid (CSF) along with an energy-sensing enzyme, AMPK [101]. AMPK is a serine/threonine kinase that functions as a master metabolic switch implicated in gluconeogenesis pathways. Compellingly, in the CSF of TBI patients, activation of AMPK attenuated the pro-inflammatory ILC1 and ILC3 and raised the frequency of regulatory IL-10+ILC2 [101]. Metformin, which is a common treatment drug for type 2 diabetes due to its effects on glucose metabolism, can activate AMPK [101]. TBI model mice treated with metformin saw increased IL-10+ILC2 and decreased ILC1 and ILC3 levels in the brain meninges, which was accompanied by improved neurological outcomes [101]. These findings suggest that glucose may be necessary for the generation and function of IL-10+ILC2 in addition to RA. It is yet to be determined whether this metabolic switch towards the glycolytic pathway and away from fatty acid metabolism contributes to the transition toward regulatory ILC2.

Furthermore, Miyamoto et al. [93] reported that the absence of the transcription factor Cbfβ or its receptor proteins Runx1 or Runx3 in ILC2, both in vitro and in vivo, increased the presence of TIGIT+IL-10+ILC2 and reduced airway inflammation. In severe airway anaphylaxis, a group of TIGIT+IL-10+ILC2 was identified as seemingly “exhausted” cells since they expressed the inhibitory receptor, TIGIT, characteristic of exhausted CD8+ T cells [2]. However, a different study concluded that IL-10 production is not because of exhausted ILC2 since Sema6d−/− mice, which had increased expression of IL-10, had a comparable expression of exhaustion markers like TIGIT with WT ILC2 [120]. As of now, it is unclear whether regulatory ILC2 are phenotypically similar to Tregs or exhausted CD8+ T cells.

It has also been reported that transcription factors cMaf and Blimp-1 can regulate IL-10 generation in several immune cells, such as CD4+ and CD8+ T cells [80,122,123]. These transcription factors have previously been implicated in suppressing inflammation [122,124]. In several experiments, IL-4 stimulation resulted in the activation of cMaf and Blimp-1 within the nucleus that regulated IL-10 production in Th1, Th2, Th17, Treg, and regulatory B (Breg) cells [121,122,123,124,125]. Recently, it has been discovered that these transcription factors are also engaged in IL-10+ILC2, as knockdown models of cMaf and Blimp-1 led to a significant decline in IL-10+ILC2 numbers [80].

## 10. Conclusions

Current literature suggests that a number of cytokines, including IL-2, IL-4, IL-10, IL-33, RA, and NMU alone or in combination, are the most promising inducers of IL-10 production by ILC2 (Figure 3). Specifically, KLRG1 and CTLA-4 surface receptors are also associated with the induction of IL-10 expression by ILC2. Additionally, ILC2-derived IL-10 can perpetuate the anti-inflammatory functions of ILC2 in an autocrine fashion. Conversely, TL1A, TGF-β, Sema6D, and the presence of surface receptor ICAM-1 have inhibitory effects on IL-10-generating ILC2. There seems to be a shift in the metabolic environment towards glycolytic pathways in vivo, further enhancing the regulatory phenotype of ILC2. Given the capacity of ILC2 for abundant secretion of cytokines relative to their cell numbers, promoting endogenous, persistent IL-10 may be a basis for targeted treatment of inflammatory conditions. In tandem, the ratio of conventional ILC2 to IL-10+ILC2 may serve as a biomarker for the evaluation of AIT efficacy. Understanding the tissue microenvironmental cues and intracellular mechanisms involved in generating regulatory ILC2 is essential for uncovering allergen tolerance and immunomodulation. Finally, recognizing the plasticity of ILC and the extent of ILC responsiveness to the local tissue milieu raises the question of whether ILC classifications require refinement.

## Figures and Tables

**Figure 1 cells-12-02556-f001:**
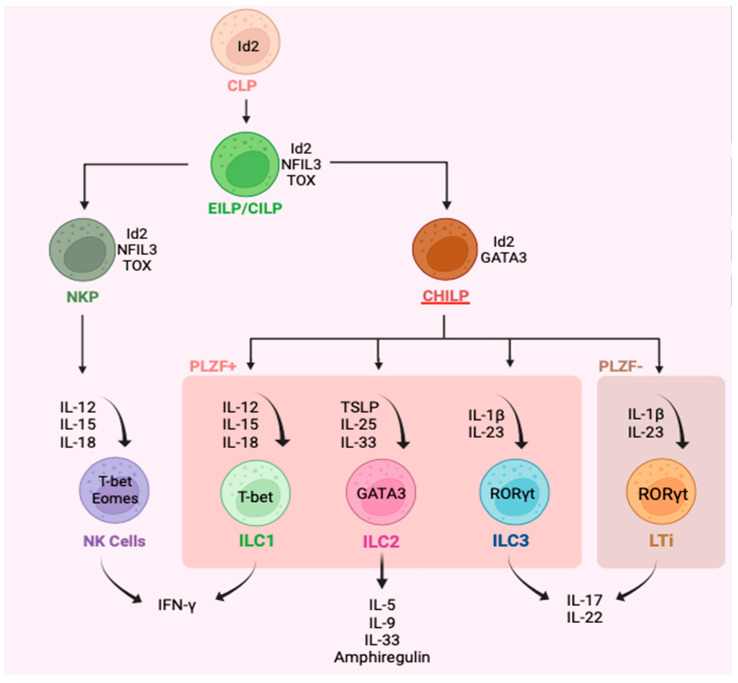
Representation of the major innate lymphoid cell types (natural killer cells, innate lymphoid types 1-3, lymphoid tissue inducers) along with their progenitor cells, key cytokine inducers, and main cytokine products, as well as identifying transcription factors. Type 1-3 innate lymphoid cells (ILC) are defined as lin−CD94−CD127+. ILC1 are generally identified as lin−CD127+CD117−CD56+/−CRTH2−NKp44−, ILC2 as lin−CD127+CD117+/−CRTH2+ and ILC3 as lin−CD56+/−CD127+CD117+CRTH2−.

**Figure 2 cells-12-02556-f002:**
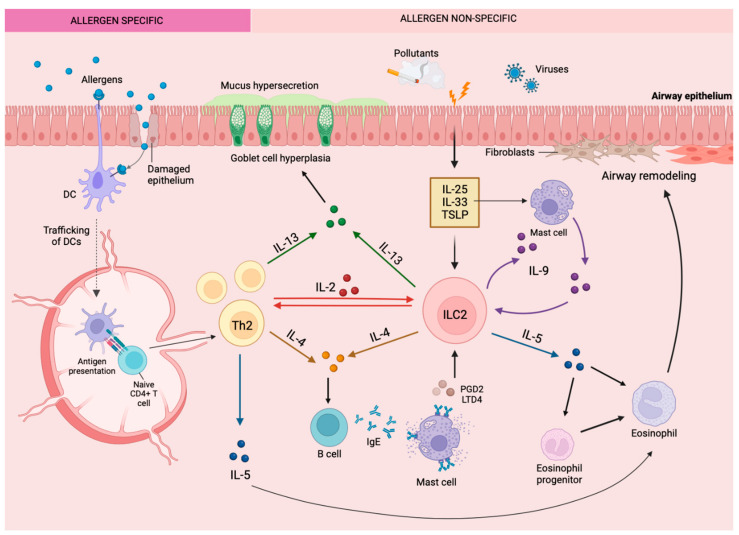
The primary role of ILC2 in the type 2 inflammatory response within the lungs in eosinophilic allergic asthma. Disruption of the bronchial epithelial barrier results in cell activation and release of alarmin cytokines that activate type 2 innate lymphoid cells, triggering the release of type 2 cytokines and the cascade of type 2 inflammatory responses characteristic of eosinophilic asthma.

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
