# Peer review of "Regulatory ILC2—Role of IL-10 Producing ILC2 in Asthma"

_cells, 2023, doi:10.3390/cells12212556_

Round 1

Reviewer 1 Report

Comments and Suggestions for Authors

In this manuscript the authors present a review of Regulatory ILC – Role of IL-10 producing ILC2 in asthma. The review presents the role of ILC2 and its role in inflammatory responses, with special focus on newly disovered regulatory subtype of ILC2, with the capacity of producing IL-10, with its regulation and consequences being described in details.

The topic is a relevant contribution in the field, the refefences are up-to-date and the role of regulatory ILC2 is clearly presented.

Besides some minor points stated bellow the the manuscript would be improved if the authors would add a paragraph on possible therapeutic options targeting regulatory ILC2.

Minor points:

1. Use italics were naming genes (e.g. IL10 in line 185 should be in italics; il10 in line 253 also,…).

2. In line 59 replace C-kit with CD117 and in line 60 remove C-kit, since CD117 is already mentioned.

3. In line 117 remove “of”.

4. In line 144 replace C-kit+ by CD117+.

5. In line 186; reference cited [79] in not about mutations of IL10. Probably the correct reference here would be [87].

6. Explain the abbreviation used; GFP in line 261.

7. In lines 310-311, the sentence “shift from a pro-inflammatory to an anti-inflammatory profile of cytokines”; probably is missing “is observed”.

8. In line 329 TFG shod be corrected to TGF.

Author Response

Reviewer 1

In this manuscript the authors present a review of Regulatory ILC – Role of IL-10 producing ILC2 in asthma. The review presents the role of ILC2 and its role in inflammatory responses, with special focus on newly discovered regulatory subtype of ILC2, with the capacity of producing IL-10, with its regulation and consequences being described in details. The topic is a relevant contribution in the field, the references are up-to-date and the role of regulatory ILC2 is clearly presented.

Besides some minor points stated bellow the manuscript would be improved if the authors would add a paragraph on possible therapeutic options targeting regulatory ILC2.

Thank you for your useful suggestions and your diligence in editing the minor errors in the text. As of yet, there are no specific treatments targeting regulatory ILC2. However, a few more sentences were added regarding therapeutic options using IL-10 on lines (231-236, 240-247, 261-271).

Minor points:

  1. Use italics were naming genes (e.g. IL10 in line 185 should be in italics; il10 in line 253 also,…).

This has now been corrected to italicize the gene names. Thank you for your diligence in catching these errors.

  1. In line 59 replace C-kit with CD117 and in line 60 remove C-kit, since CD117 is already mentioned.

This has now been corrected to reflect the suggested changes. We appreciate your attention to these important details.

  1. In line 117 remove “of”.

This typo has been corrected accordingly.

  1. In line 144 replace C-kit+ by CD117+.

            Replacement is complete. Again, thank you for identifying these.

  1. In line 186; reference cited [79] in not about mutations of IL10. Probably the correct reference here would be [87].

This must have occurred due to a glitch when changing our reference manager, so we are grateful for your vigilance in detecting these important inaccuracies. The reference has now been fixed on lines (198).

  1. Explain the abbreviation used; GFP in line 261.

In accordance with your thoughtful suggestion, a brief explanation has been added to lines 298 and 301.

  1. In lines 310-311, the sentence “shift from a pro-inflammatory to an anti-inflammatory profile of cytokines”; probably is missing “is observed”.

The sentence has been altered to reflect this revision.

  1. In line 329 TFG shod be corrected to TGF.

The typo has been corrected.

Reviewer 2 Report

Comments and Suggestions for Authors

The authors summarize recent findings on IL-10-producing ILC2, including that ILC2 expresses KLRG1, produces IL-10 upon stimulation with IL-2, IL-4, or RA, and is suppressed by TL1A; that this IL-10-producing ILC2 is increased in responders to allergen immune therapy; and that allergic diseases such as asthma are not exacerbated when these cells increase. There are some inaccuracies in descriptions and sources that need to be corrected.

Major comments

The authors state in Line 62 that NK cell is included in ILC, but Line 56 contradicts this by stating that cells with a negative for CD94, a NK marker, are ILC.

Line 110; The authors state that ILC2 produces 500 times more T2 cytokines than CD4+Th2 cell, but in reference 20 it is 100-fold, and I could not find such data in references 8 and 56. Please check the numbers and/or the references.

Line 278-280; The authors write “Interestingly, mice that were IL-2 deficient did not show this effect, speaking to the importance of IL-2 signaling in promoting regulation and attenuation of anti-inflammatory cells [98]” but reference 98 do not include such data. Baban et al. used IL-2RgammaKO but not IL-2KO mice.

Line 284-285: There is no data in the reference 95 that mast cell produces RA. In reference 79 it is written that CD103+DC produces RA. ”Mast cells and CD103+ dendritic cells (DC) have been described as in vivo sources of IL-2 and RA, respectively.”

Line 302; “Th2-derived IL-4 is also necessary for the accumulation of inflammatory ILC2 within the lungs” The reviewer could not find the source of this statement because  reference 99 didn't cover anything about ILCs, so please confirm that your references are correct.

Line 340-342; Reference 81 mentions nothing about E-cadherin. Is there any other ligand for KLRG1 than E-cadherin?

Line 353-368; The involvement of KLRG1+ILC2 in sex differences is shown in references 109-113, but these papers do not seem to include any data indicating IL-10 production from ILC2. Is IL-10 from ILC2 surely involved as a cause of sex differences?

Line 392-393; Isn't cytokine production altered in both Sema6dKO and WT, but in KO compared to WT?

Minor comments

The full spelling of TSLP is given on line 88, but TSLP appears first on line 50, so please provide the full spelling here.

Line 282, The authors probably mistook the reference they cite for another paper by the same first author.

Line 325; Reference 105 does not mention TL1A/DR3. Please confirm the citation.

Line 329; Correct TFG to TGF.

Line 389; Correct Semaphoring to Semaphorin 6D.

The location of Figure 3 and table 1 are far from the corresponding location of the text, making it difficult to read.

Author Response

Reviewer 2

The authors summarize recent findings on IL-10-producing ILC2, including that ILC2 expresses KLRG1, produces IL-10 upon stimulation with IL-2, IL-4, or RA, and is suppressed by TL1A; that this IL-10-producing ILC2 is increased in responders to allergen immune therapy; and that allergic diseases such as asthma are not exacerbated when these cells increase. There are some inaccuracies in descriptions and sources that need to be corrected.

 Major comments

The authors state in Line 62 that NK cell is included in ILC, but Line 56 contradicts this by stating that cells with a negative for CD94, a NK marker, are ILC.

Many thanks for this critical observation. In fact, the expression of CD94 on NK cells, and the lack of it on ILC1, is one of the distinguishing factors between these two cells (Spits et al., 2016). Meanwhile, all ILC including NK cells are lineage negative cells, meaning they do not express the lineage markers being CD3, CD14, CD16, CD19, CD20, CD56, and FcER1, that distinguish them from adaptive immune cells such as T and B cells. It has now been clarified within the text that it is only the group 1-3 ILC that are CD94-CD45+CD127+ cells on line 56. We apologize for any misunderstanding.

https://doi.org/10.1038/ni.3482

Line 110; The authors state that ILC2 produces 500 times more T2 cytokines than CD4+Th2 cell, but in reference 20 it is 100-fold, and I could not find such data in references 8 and 56. Please check the numbers and/or the references.

Apologies for the mix up, we have corrected the manuscript to correctly reflect a 100-fold difference in cytokine production.

Line 278-280; The authors write “Interestingly, mice that were IL-2 deficient did not show this effect, speaking to the importance of IL-2 signaling in promoting regulation and attenuation of anti-inflammatory cells [98]” but reference 98 do not include such data. Baban et al. used IL-2RgammaKO but not IL-2KO mice.

This is completely true, and we thank you for your diligence in spotting this inconsistency. We have now corrected the writing to reflect that IL-2 receptor gamma was knocked out, not the IL-2 cytokine itself, as seen on lines 316, 317.

Line 284-285: There is no data in the reference 95 that mast cell produces RA. In reference 79 it is written that CD103+DC produces RA. “Mast cells and CD103+ dendritic cells (DC) have been described as in vivo sources of IL-2 and RA, respectively.”

We appreciate you bringing our attention to this. The wording has been altered to correctly reflect that mast cells are a source of IL-2 and not retinoic acid on lines 325-328.

Line 302; “Th2-derived IL-4 is also necessary for the accumulation of inflammatory ILC2 within the lungs” The reviewer could not find the source of this statement because reference 99 didn't cover anything about ILCs, so please confirm that your references are correct.

It seems that the reference was missed for this information, and we are grateful for the time and care you have put into providing these very useful edits. The reference has now been updated to reflect the proper source showing Th2-derived IL-4 is necessary for accumulation of inflammatory ILC2, as can be seen on lines 344-346.

https://doi.org/10.1002/eji.201948161

Line 340-342; Reference 81 mentions nothing about E-cadherin. Is there any other ligand for KLRG1 than E-cadherin?

Reference 81 only characterized cells based on whether they express KLRG1 and what cytokines they produce, but they did not look at stimulation of KLRG1 through ligands. However, for the sake of added context and speculation regarding the possible pathways involved in KLRG1+ILC2, we have mentioned that E-cadherin is generally the main known ligand for KLRG1. Nevertheless, your observation is completely valid that this referencing can be misleading, therefore, this sentence has now been removed.  

Line 353-368; The involvement of KLRG1+ILC2 in sex differences is shown in references 109-113, but these papers do not seem to include any data indicating IL-10 production from ILC2. Is IL-10 from ILC2 surely involved as a cause of sex differences?

The purpose of the section was to speak on the need for investigation into the association between IL-10+ILC2 and sex differences in asthma. So far, IL-10 from ILC2 has not been shown to be involved in sex differences. However as mentioned, KLRG1 expression on ILC2 is associated with sex differences, in that males have more KLRG1 expression on ILC2 and tend to experience less severe illness. Separately, IL-10 production by ILC2 has been associated with KLRG1 expression on these cells (Reference 81). Therefore, the authors were speculating whether there could possibly be a connection between KLRG1 expression, IL-10+ILC2, and sex differences, which remains to be investigated. Please accept our sincere apologies if this was not initially clear in the writing. In line with your comment, this section (lines 396 and 410-413) was altered to further clarify this point within the paper.

Line 392-393; Isn't cytokine production altered in both Sema6dKO and WT, but in KO compared to WT?

You are absolutely correct that cytokine production was altered significantly when comparing KO and WT mice. The wording on lines 454-456 has been altered to appropriately reflect this. Thank you for bringing this to our attention so that we could rectify the writing.  

Minor comments

The full spelling of TSLP is given on line 88, but TSLP appears first on line 50, so please provide the full spelling here.

Thank you for your vigilance. We have now added the full spelling of TSLP to the first time it appears in the text on line 50.

Line 282, The authors probably mistook the reference they cite for another paper by the same first author.

We really appreciate you pointing this out. Here, the authors were trying to speak on the possibility that IL-2 and RA contribute to the effect that IL-33 has on IL-10+ILC2, as a part of speculation about mechanisms of action of these cytokines. The first reference is the review by Sun et al. which states that IL-2 and RA can increase the efficacy of IL-33 to induce IL-10. Meanwhile, the second reference, which is the 2015 Morita et al. shows how IL-2 is produced by mast cells as a result of IL-33 stimulation, showing a possible mechanism by which IL-2 can act as a proxy or amplifier of the effects of IL-33. For this reason, the authors included this reference in this sentence. However, the 2019 paper by Morita et al. looks at the impact of in vitro stimulation of ILC2 with different combinations of IL-2, IL-33, and RA, and does not make any claims regarding how these cytokines interact or impact each other.    

Line 325; Reference 105 does not mention TL1A/DR3. Please confirm the citation.

Reference 105 has been removed from the citation on this line. We continue to be appreciative of your diligence in catching these minor inconsistencies.

Line 329; Correct TFG to TGF.

This typo has been corrected in the text.

Line 389; Correct Semaphoring to Semaphorin 6D.

This typo has also been corrected in the text.

The location of Figure 3 and table 1 are far from the corresponding location of the text, making it difficult to read.

Your feedback in this regard is much appreciated. In accordance, Figure 3 has been moved to show up at the end of the relevant section, being “Positive Inducers of IL-10 Production by ILC2” section. Table 1 has also been moved up to appear right at the end of the “Inhibitors of IL-10 Expression by ILC2” section.

Round 2

Reviewer 2 Report

Comments and Suggestions for Authors

The authors misunderstood about IL-2Rg KO (but not IL-2 KO) mice.  There is have no ILC2 cells in IL-2Rg KO mice. That's why it cannot be judged that IL-2 signaling is required for IL-10 production. Please revise the context of the text (around line 300).
Furthermore, after the addition of new references, the order of references seem to be in wrong location of the text. The author should check them, carefully (line 226-307).

Author Response

Reviewer Comments

  1. The authors misunderstood about IL-2Rg KO (but not IL-2 KO) mice.  There is have no ILC2 cells in IL-2Rg KO mice. That's why it cannot be judged that IL-2 signaling is required for IL-10 production. Please revise the context of the text (around line 300).

Thank you so much for clarifying. Here, the authors were speaking on the involvement of the IL-2 pathway in inducing IL-10 production generally, and not just within ILC2 since this paper looks at Treg cells. However, to avoid any confusion or misunderstanding, we have removed this sentence from the text.

  1. Furthermore, after the addition of new references, the order of references seem to be in wrong location of the text. The author should check them, carefully (line 226-307).

We really appreciate your diligence in catching this. All references have been fixed throughout the text.